Submitted to the Proceedings of the US Community Study
on the Future of Particle Physics (Snowmass 2021)

# Snowmass2021 Cosmic Frontier White Paper:
# Ultraheavy particle dark matter

Daniel Carney,[1, *] Nirmal Raj,[2, 3, †] Yang Bai,[4] Joshua Berger,[5] Carlos Blanco,[6, 7] Joseph
Bramante,[8, 9] Christopher Cappiello,[8] Thomas Y Chen,[10] Maíra Dutra,[11] Reza Ebadi,[12, 13] Kristi
Engel,[12] Edward Kolb,[14] J. Patrick Harding,[15] Jason Kumar,[16] Gordan Krnjaic,[17, 18] Rafael F.
Lang,[19] Rebecca K. Leane,[20, 21] Benjamin V. Lehmann,[22, 23] Shengchao Li,[19] Andrew J. Long,[24]
Gopolang Mohlabeng,[8, 9] Ibles Olcina,[1, 25] Elisa Pueschel,[26] Nicholas L. Rodd,[27] Carsten Rott,[28, 29]
Dipan Sengupta,[30, 31] Bibhushan Shakya,[32] Ronald L. Walsworth,[12, 13] and Shawn Westerdale[33]

[1] Physics Division, Lawrence Berkeley National Lab, Berkeley, CA, USA
[2] TRIUMF, Vancouver, BC, Canada
[3] The Arthur B. McDonald Canadian Astroparticle Physics Research Institute, Kingston, ON, Canada
[4] Department of Physics, University of Wisconsin-Madison, Madison, WI, USA
[5] Department of Physics, Colorado State University, Fort Collins, CO, USA
[6] Department of Physics, Princeton University, Princeton, NJ, USA
[7] The Oskar Klein Centre, Stockholm University, AlbaNova, Stockholm, Sweden
[8] The Arthur B. McDonald Canadian Astroparticle Physics Research Institute and Department of Physics,
Engineering Physics and Astronomy, Queens University, Kingston, Ontario, Canada
[9] Perimeter Institute for Theoretical Physics, Waterloo, Ontario, Canada
[10] Fu Foundation School of Engineering and Applied Science,
Columbia University, New York, NY, USA
[11] Ottawa-Carleton Institute for Physics, Carleton University, Ottawa, Canada
[12] Department of Physics, University of Maryland, College Park, MD, USA
[13] Quantum Technology Center, University of Maryland, College Park, MD, USA
[14] Kavli Institute for Cosmological Physics and Enrico Fermi Institute,
The University of Chicago, Chicago, IL, USA
[15] Physics Division, Los Alamos National Laboratory, Los Alamos, NM, USA
[16] Department of Physics & Astronomy, University of Hawai'i, Honolulu, HI, USA
[17] Theoretical Physics Department, Fermilab, Batavia Il, USA
[18] Kavli Institute for Cosmological Physics,
University of Chicago, Chicago, IL, USA
[19] Department of Physics, Purdue University, West Lafayette, IN USA
[20] SLAC National Accelerator Laboratory, Stanford University, Stanford, CA, USA
[21] Kavli Institute for Particle Astrophysics and Cosmology, Stanford University, Stanford, CA, USA
[22] Department of Physics, University of California, Santa Cruz, CA, USA
[23] Santa Cruz Institute for Particle Physics, Santa Cruz, CA, USA
[24] Department of Physics and Astronomy, Rice University, Houston, TX, USA
[25] University of California, Berkeley, Department of Physics, Berkeley, CA, USA
[26] Deutsches Elektronen-Synchrotron DESY, Zeuthen, Germany
[27] Theoretical Physics Department, CERN, Geneva, Switzerland
[28] Department of Physics, Sungkyunkwan University, Suwon, Korea
[29] Department of Physics and Astronomy,
University of Utah, Salt Lake City, UT, USA

$^{30}$*Department of Physics and Astronomy,*
*University of California, San Diego, USA*
$^{31}$*ARC Center of Excellence for Particle Physics, University of Adelaide, Australia*
$^{32}$*Deutsches Elektronen-Synchrotron DESY, Hamburg, Germany*
$^{33}$*Department of Physics, Princeton University, Princeton, USA*

We outline the unique opportunities and challenges in the search for "ultraheavy" dark matter candidates with masses between roughly 10 TeV and the Planck scale $m_{\rm pl} \approx 10^{16}$ TeV. This mass range presents a wide and relatively unexplored dark matter parameter space, with a rich space of possible models and cosmic histories. We emphasize that both current detectors and new, targeted search techniques, via both direct and indirect detection, are poised to contribute to searches for ultraheavy particle dark matter in the coming decade. We highlight the need for new developments in this space, including new analyses of current and imminent direct and indirect experiments targeting ultraheavy dark matter and development of new, ultra-sensitive detector technologies like next-generation liquid noble detectors, neutrino experiments, and specialized quantum sensing techniques.

## CONTENTS

## I. INTRODUCTION

*Contributors: Daniel Carney, Edward Kolb*

For decades, the search for dark matter (DM) has focused on two mass regions: ultralight axions (or axion-like particles) with mass $m_\chi \lesssim 1$ eV, and particles with mass around the electroweak scale $m_\chi \approx 100$ GeV with weak scale couplings (WIMP). Unfortunately, in spite of heroic experimental efforts, ultralight or weak-scale DM particles have not been found. This has motivated theorists to propose new cosmological mechanisms for the production of DM, and for experimentalists to study new ways to search in unexplored ranges of mass and interaction strength.

This community white paper focuses on one such less-explored region of parameter space: "ultraheavy" dark matter (UHDM). Here by ultraheavy we will mean particles that have a mass too large to be produced at any current colliders, $m_\chi \gtrsim 10$ TeV, while also having mass below roughly the Planck mass $m_\chi \lesssim m_{\rm pl} \approx 10^{19}$ GeV. The lower bound also roughly corresponds to the traditional relic production threshold $m_\chi \gtrsim 100$ TeV set by unitarity bounds on the production cross-section (see Sec. II for an extended discussion). The upper bound comes from considering the expected

* carney@lbl.gov
† ep05b010@gmail.com

number density of DM: under standard halo density assumptions, Planck-scale dark matter would produce a flux of order 1 event/m$^2$/yr [see Eq. (1)]. Thus dark matter much beyond the Planck mass would be very difficult to detect directly in a terrestrial experiment. The UHDM parameter space constitutes a vast and relatively unexplored frontier, and our aim in this whitepaper is to outline the unique challenges and opportunities it presents.

We begin with an overview of the cosmological history and theory of such DM candidates. In this mass regime, new production mechanisms beyond the usual cold thermal relic picture must come into play. In addition to fundamental particles, a diverse set of DM candidates becomes viable, including composite objects, solitons, and relics of decaying black holes.

We then move on to a discussion of detection prospects, aiming to motivate further work with new and existing detection techniques. In the direct DM detection program, we emphasize that current and next-generation detectors built to search for usual DM candidates are also well-placed to search for certain UHDM candidates. We also highlight some ideas for future experiments aiming specifically for heavier DM detection. Finally, we emphasize the possibility of indirect detection of UHDM unstable to decays to visible signatures with a variety of current and upcoming observatories.

## II.   COSMIC HISTORY AND MODELS

*Contributors: Yang Bai, Joseph Bramante, Maíra Dutra, Gopolang Mohlabeng, Ben Lehmann, Andrew Long, Dipan Sengupta, Bibhushan Shakya, Carlos Blanco*

While much is known about the synthesis of Standard Model states, starting with Big Bang Nucleosynthesis (BBN) onward to the present era of galaxies and accelerated cosmic expansion, very little is known about the state of the universe prior to BBN, when the universe had a temperature of around a few MeV. The synthesis of DM particles is usually presumed to occur before BBN, and the synthesis of UHDM particles in particular is highly dependent on the unknown physics of the early universe. In this section, we highlight a variety of cosmological scenarios and models of UHDM leading to the observed DM abundance.

The emergence of a concordance $\Lambda$ Cold Dark Matter ($\Lambda$CDM) cosmology points to evidence that Standard Model particles were once thermalized in the early universe [1]. Currently, we can empirically infer the thermal history of the universe back up to the BBN scale, when ultra-relativistic species (*radiation*) dominated the cosmic expansion. However, one explanation for the flat, homogeneous and isotropic state of the present universe is that it has undergone a phase of exponential expansion, i.e. inflation. The inflaton field driving inflation generates the radiation content, and therefore the cosmic entropy, via out-of-equilibrium decay [2]. This so-called *reheating* period dilutes any cosmic relic, which leads one to expect that DM was produced after inflation.

It is an open question whether the universe was always radiation-dominated from the end of inflationary reheating or reheating after some other antecedent period up to the epoch of BBN. For example, one simple possibility is that an exotic BSM field fell out of equilibrium with a reheated SM bath, and grew to dominate the cosmic expansion after becoming non-relativistic, leading to a period of *early matter domination* (EMD) prior to BBN. In this case, the universe would usually have undergone a *late reheating* period associated with the decay of this new field, injecting considerable entropy into the cosmic bath and diluting the abundance of all decoupled particles. As a consequence, many DM models predicated on a purely radiation-dominated universe will entail heavier dark sector states, to compensate for late time dilution. On the other hand, EMD also provides an intriguingly simple DM production scenario: the BSM field could decay directly to a heavy, out-of-equilibrium DM state. This process alone could set the relic abundance of DM.

## production of ultra-heavy dark matter

FIG. 1. A non-exhaustive representation of cosmological production mechanisms of ultraheavy dark matter and corresponding models.

The abundance can be determined, as described below, via several mechanisms that may have occurred during eras of radiation, matter, or vacuum energy domination. In what follows, we describe each of these production mechanisms. These are summarized in Figure 1.

**Freeze-out:** In this classic mechanism, DM particles begin in thermal equilibrium with the SM bath, with equal rates of DM production and annihilation. When DM particles become non-relativistic, their production is Boltzmann-suppressed, and they fall out of equilibrium as cosmic expansion becomes faster than annihilation.

Partial-wave unitarity sets an upper limit on perturbative DM annihilation cross sections. When DM is produced via freeze-out in a radiation-dominated universe, an upper limit on $s$-wave $2 \to 2$ annihilation cross sections leads to a lower limit on the DM abundance, in turn translating to an upper limit of about 100 TeV on the DM mass [3]. However, if an EMD occurred after freeze-out, smaller annihilation cross sections would be needed to overcome the dilution and lead to the correct amount of DM. In this case, frozen-out DM with masses beyond $\mathcal{O}(100\ \text{TeV})$ become allowed [4–6]. In models in which DM is much heavier than mediators, Sommerfeld enhancement of cross sections and the formation of bound states alters unitarity bounds [7–11]. DM might also be part of a hidden thermal bath, with a temperature different from the SM [12]. The lightest particles of the hidden sector can dominate the cosmic expansion after freeze-out, leading to an EMD era. In this case, *diluted* DM particles as heavy as $10^{10}$ GeV become viable [13–16]. The unitarity bound can also be circumvented in scenarios with additional degrees of freedom in the dark sector. In particular, in the presence of an additional species $\zeta$ with $m_\zeta < m_\chi < 3m_\zeta$, the DM $\chi$ remains stable, but the process $\chi\zeta^\dagger \to \chi\chi$ attenuates the relic density and allows $m_\chi$ to be as large as $10^9$ GeV [17]. If the dark sector consists of many nearly-degenerate species that scatter with the SM through dark-flavor–changing interactions, $m_\chi$ can be as large as $10^{14}$ GeV without violating the unitarity bound [18].

**Freeze-in:** In the freeze-in mechanism, the DM population is initially negligible, and is produced via out-of-equilibrium decays and/or annihilation of species in the SM bath [19–21]. The

production rates are always slower than the cosmic expansion and become negligible before back-reaction becomes important. The end of the freeze-in production depends on DM-SM interactions. Typically, renormalizable couplings lead to an *infrared* freeze-in, in which DM production stops when it becomes too heavy to be produced and the final relic density depends only on the DM coupling strengths and mass [1]. On the other hand, non-renormalizable couplings typically lead to production rates with a high temperature-dependence. In this case, freeze-in can terminate during the post-inflationary reheating and is said to be *ultraviolet*, with a final relic density depending on the reheat temperature. DM candidates produced via ultraviolet freeze-in (UVFI) only need to be lighter than the maximal temperature of the SM bath, which can be as high as $10^{15}$ GeV [23–25]. In fact, the earliest proposals of UVFI detailed the production of UHDM candidates [26–28].

**Out-of-equilibrium decay:** DM can be directly produced from the decay of heavy fields which are not part of a thermal bath. This is the case of inflaton [29–31] and moduli [32–34] fields. Even when DM is heavier than the inflaton field and cannot be produced via decay during the inflationary reheating, nonperturbative quantum effects at the onset of inflaton oscillation (*preheating*) can still produce UHDM [35–37], with mass at the GUT scale [38]. Preheating could also produce **dark monopole** states that constitute DM [39]. It is also worth mentioning that the highly energetic decay products of heavy fields might also produce UHDM particles before the thermalization process is complete [40].

**Phase transitions:** UHDM can be produced at various stages of a first order phase transition, and can also be accompanied by gravitational wave signals. Production of UHDM with masses much higher than the energy scale of the phase transition can occur when particles present in the plasma cross ultrarelativistic bubble walls [41], or when such ultrarelativistic bubble walls collide [42]. UHDM can also obtain the correct relic density when the phase transition occurs in a confining sector and is supercooled, resulting in an appropriate dilution of the DM abundance [11, 43]. Bubble walls can filter dark matter particles out of the plasma and thereby control their relic abundance [44, 45] or collect them into composite objects [46, 47]. A first order electroweak phase transition could produce DM in the form of **electroweak-symmetric solitons** [48].

**Gravitational particle production:** All of the observational evidence for DM in our universe arises from its gravitational interactions with ordinary states. Many DM models assume additional non-gravitational interactions, but explain DM production via gravity in the early universe.

The phenomenon of inflationary gravitational particle production [49, 50], which occurs for quantum field theories in curved spacetime [51, 52], can explain DM production, e.g. **WIMPzillas** [27, 53–55]. Typically, although though there are important exceptions, production is most efficient when the DM mass is comparable to the inflationary Hubble scale, $m_\chi \sim H_{\rm inf}$, and since cosmological observations constrain $H_{\rm inf} \lesssim 10^{14}$ GeV, such models usually involve superheavy elementary particles. Notable recent work has explored models of higher-spin DM [56–61], models of superheavy DM with $m_\chi > H_{\rm inf}$ [62–64], improved analytical techniques [65–67], and cosmological signatures such as isocurvature [68–70].

**Primordial (extremal) black holes:** Primordial black holes (PBHs) are a well-known DM candidate, and heavy PBHs are best regarded as compact objects for phenomenological purposes. However, very light PBHs have much more in common with elementary particles than with astrophysical compact objects. In particular, a wide class of models accommodate non-evaporating black hole remnants at the Planck scale or below. Such objects are effectively ultraheavy particles, and have long been considered a viable DM candidate [71, 72]. This scenario provides a natural formation mechanism for a secluded dark sector: any sufficiently light population of PBHs quickly

---

[1] One can engineer complicated models of IR freeze-in with significantly heavy DM candidates, viz., the clockwork scenarios [22].

evaporates, leaving remnants that may only interact gravitationally. Without any BSM fields, magnetic black hole solutions with a "hairy" cloud of electroweak gauge and Higgs fields exist [73–76]. Depending on the UV completion for quantum gravity, the ***Planck-scale relics*** of the black-hole-like endpoints of gravitational collapse may be regarded as a general DM candidate, e.g. [77]. These remnants could carry $U(1)$ charges which could lead to unique observational signatures [78, 79].

## III.   DIRECT DETECTION

*Contributors: Yang Bai, Joseph Bramante, Christopher Cappiello, Daniel Carney, Reza Ebadi, Jason Kumar, Rafael Lang, Nirmal Raj, Ibles Olcina, Shawn Westerdale*

Searches for ultraheavy dark matter are also possible with direct detection experiments, including currently existing detectors designed to look for much lighter DM candidates. In this section, we aim to motivate the basic detection problems and methods to perform searches both with existing and purpose-built future experiments.

As the mass of each DM constituent increases, the number density and corresponding flux of particles decreases. Assuming the standard DM mass density of $\rho \approx 0.3$ GeV/cm$^3$ and mean velocity of $\overline{v} \approx 220$ km/s, the expected flux of DM particles of individual mass $m_\chi$ passing through a detector is

$$\Phi = n\overline{v} \approx \frac{0.85}{\text{m}^2 \text{ yr}} \times \left( \frac{m_{\text{pl}}}{m_\chi} \right). \tag{1}$$

Even a background-free detector is limited by needing at least a handful of dark matter particles to pass though during its lifetime. In a typical single-scatter search, the sensitivity of an experiment scales with the fiducial mass of the detector. However, ultraheavy dark matter may scatter several times as it crosses the detector for large enough cross sections, in which case the area of the detector becomes the most relevant factor [80].

The detailed sensitivity of a DM experiment to UHDM depends on how the UHDM couples to the Standard Model constituents in the detector. As examples of the basic ideas, here we will focus on two key cases. The first is UHDM coupled to nuclei through a weak, short-range interaction; essentially a much heavier version of the WIMP. The second is UHDM coupled to the Standard Model for a long-range force, which in the ultimate limit could simply be the gravitational coupling.

Consider first a search for UHDM with a weak contact interaction with nuclei. Fig. 2 shows current limits and projections on the ultraheavy parameter space under two models with different relationships between the DM-nucleus ($\sigma_{T\chi}$) and DM-nucleon ($\sigma_{n\chi}$) cross sections, respectively. In Model I (left) DM is opaque to the nucleus and no scaling from nucleon to nucleus is assumed, while in Model II (right) the typical $A^4$ scaling arising from contact interactions with the Born approximation is assumed. For more details on these models see Refs. [81, 99]. We note that when the DM-nucleus cross section approaches the geometric cross section of the nucleus, the Born approximation is no longer valid, and this $A^4$ scaling breaks down. The breakdown leads to a cross section that saturates at this geometric limit for a repulsive interaction, and one which displays resonant behavior for an attractive interaction. However, it may be possible to preserve the $A^4$ scaling for models of composite DM or light mediators.

It is important to note that for a large enough scattering cross section, dark matter can scatter so often in the overburden and lose enough energy that it becomes undetectable when it reaches an experiment, depending on the detector's energy threshold [80, 97]. For relatively light DM, this attenuation is most accurately modeled using Monte Carlo simulations, which can track the trajectories of individual particles in 3-dimensional space as they scatter with nuclei in the overburden

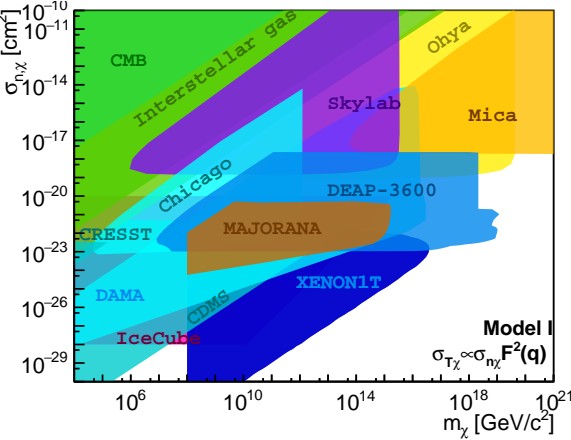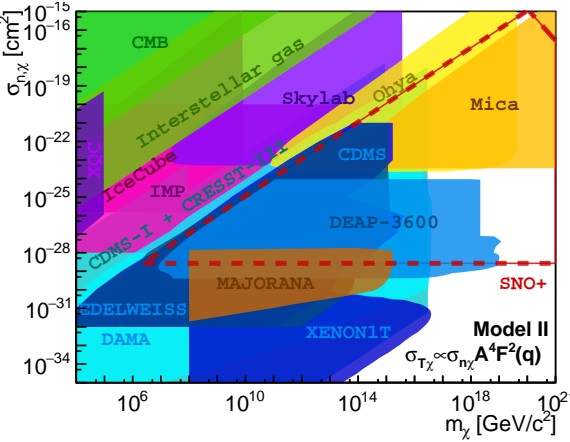

FIG. 2. Current and projected experimental regions of ultraheavy parameter space excluded by cosmological/astrophysical constraints (green), direct detection dark matter detectors (blue), neutrino experiments (red/orange), space-based experiments (purple), and terrestrial track-based observations (yellow). Both models considered here assume different relations for the cross section scaling from a single nucleon to a nucleus with mass number $A$. In the left plot, we assume no scaling with $A$; in the right plot, we assume the cross section scales like $A^4$ (e.g., with two powers coming from nuclear coherence, and two from kinematic factors). Limits are shown from DEAP-3600 [81], DAMA [82, 83], interstellar gas clouds [84, 85], a recast of CRESST and CDMS-I [86], a recast of CDMS and EDELWEISS [87, 88], a detector in U. Chicago [89], a XENON1T single-scatter analysis [90], tracks in the Skylab and Ohya plastic etch detectors [83], in ancient mica [91], the MAJORANA demonstrator [90], IceCube with 22 strings [92], XQC [93], CMB measurements [94, 95], and IMP [96]. Also shown is the future reach of the liquid scintillator detector SNO+ as estimated in [97, 98].

(see e.g. Ref. [100]). However, when the DM is much heavier than a nucleus, it follows a nearly straight trajectory, and requires a large number of collisions to be appreciably slowed. This means that attenuation of UHDM can be modeled as a continuous energy-loss process along a straight trajectory, which is computationally much faster than the full Monte Carlo approach [86, 101]. The resulting "ceiling", the maximum cross section to which an experiment is sensitive, is approximately proportional to the DM mass for UHDM, as can be seen in many of the exclusion regions in Fig. 2.

Since the energy deposited by each interaction is independent of the DM mass (because $m_\chi \gg m_N$), the total amount of energy deposited in the detector by a passing DM particle scales linearly with the cross section. This means that the total signal can span many orders of magnitude in deposited energy, and the signal shape can vary from one to many continuous hits inside the detector. As such, it is necessary for analyses to maintain sensitivity over a large range. The broad range of possible signal manifestations highlights the need for many different detector technologies capable of sampling various regions of this model space. Other challenges that may arise in these DM searches include computational difficulties related to performing a full optical simulation with a very large number of scatters, designing DAQ schema and low-level analyses that adequately differentiate between bright, long-duration pulses produced by ultraheavy DM and instrumental noise, and, at the lower end of the multi-scatter regime, adequately discriminate against pile-up or multi-scatter backgrounds.

Since the UHDM is very heavy, it is not appreciably deflected during each scattering event, and so the signal here is essentially a track of sub-threshold events. For a detector using a time projection chamber (TPC) which is able to determine both the time and location of the energy depositions, this is a striking signal. Not only is this type of search almost background-free, but

it constitutes a direct measurement of the dark matter velocity vector [98]. With the observation of a sufficient number of such tracks, one could obtain a direct measurement of the dark matter velocity distribution. Note that this would be very different from the more commonly considered case of directional dark matter direct detection, in which one attempts to measure the energy and direction of the recoiling nucleus, and from this infer the velocity of the incoming dark matter particle. Instead, for multiply-interacting dark matter, one would directly measure the dark matter particle's velocity vector.

This track-like signature will also appear for models of UHDM coupled to the Standard Model through long-range forces, our second case study. If the force has sufficiently long range (i.e. is mediated by a sufficiently light boson), the DM will act coherently across an entire many-body target. One can then consider detecting it by using large, even macroscopic, targets [102–108]. A proof of concept demonstration was given in Ref. [107], which used a microgram-scale mechanical accelerometer to search for heavy, composite DM coupled to nuclei through a light gauged $B$-$L$ vector boson. Any new long-range force coupling to nuclei is up against strong limits from existing experiments, but current generation sensors can already go beyond these limits in certain cases [109, 110]. Ultimately, with sufficiently heavy DM, it has been suggested [102–104] that one could use the only coupling DM is guaranteed to have—gravity—to perform searches this way. This would require a large array of devices, operating in a deeply quantum regime, as pursued by the Windchime collaboration [111]. Such an array would be sensitive to a wide variety of UHDM candidates [108]. In addition, well-characterized geologically old rock samples can also serve as UHDM detectors leveraging the long track-like signature as a background discrimination tool. Samples that have been stable for more than $\sim 10^9$ years provide sensitivity to even heavier UHDM candidates due to their long exposure time [112].

We also note that if an $O(1)$ fraction of DM is composed of charged PBH remnants, as proposed by Ref. [79], such objects can be detected terrestrially by several means. In particular, these objects exhibit unique signatures in large-volume LAr detectors, and are robustly detectable given an appropriate triggering mechanism. (See also Ref. [113].) Paleo-detectors [114] are also expected to be sensitive to charged remnants due to their long exposure times. Additionally, some UHDM candidates around the Planck mass, such as electroweak-symmetric solitons, could be discovered by nuclear capture signals [115]. For a wide range of UHDM models with a geometric interaction with the target nuclei, the O(1 GeV) photon energy from the radiative capture process may also be detectable at the IceCube detectors, similar to the search for non-relativistic magnetic monopoles [116]. Moreover, certain models of composite DM that cause nuclei and leptons to be accelerated in their binding potential, result in high energy bremsstrahlung photons [117] and low energy Migdal effect electrons [118], detectable at large-volume and DD experiments. Finally, scenarios with baryon charged multiply-interacting particles coupled to low mass mediators could also be detected with liquid scintillators [119].

In conclusion, the multi-scatter frontier opens up new parameter space to be explored by direct detection and neutrino experiments. It has already been demonstrated that dedicated analyses of existing data can be very fruitful in exploring new parameter space [81, 82, 90]. As the total integrated DM fluxes of future DM experiments are expected to increase by orders of magnitudes over their run-time, the maximum DM mass reachable in a direct search experiment (e.g. DARWIN/G3, Argo, or eventually possibly ktonne-scale detectors [120]) will be able to reach beyond the Planck mass $\simeq 10^{19}\,\mathrm{GeV}/c^2$ [80]. Moreover, a variety of new detector technologies, including mechanical sensors, can be brought to bear on this frontier. These are exciting prospects for the search of ultraheavy DM and these efforts will hopefully continue to expand as larger detectors come online in the following years.

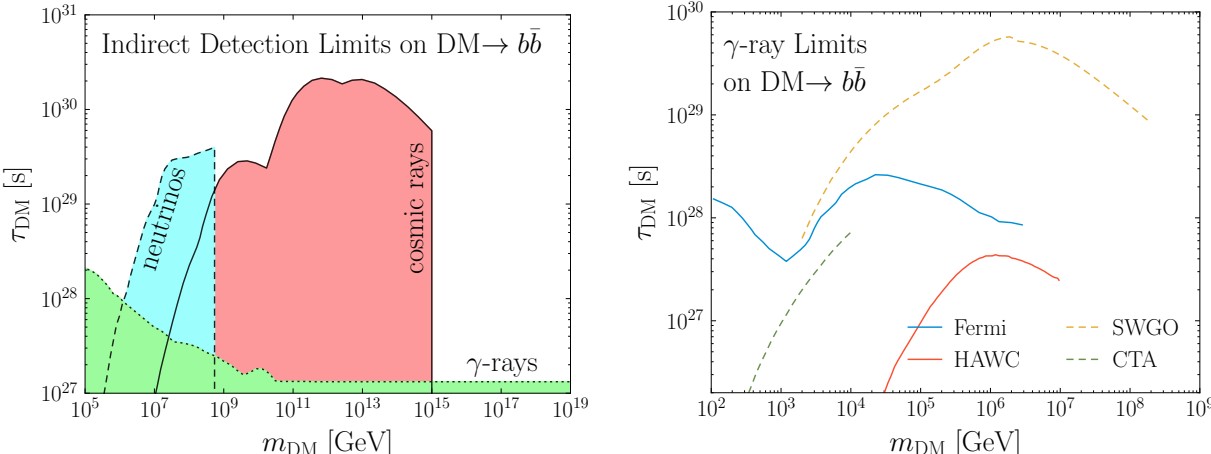

FIG. 3. (Left) A subset of current indirect detection constraints on the DM lifetime for decays to $b\bar{b}$. The results are chosen to highlight the complementarity between different search strategies for this single DM hypothesis, and in particular we show limits obtained using $\gamma$-ray [125], neutrino [126], and cosmic-ray [127] studies. See the text for additional details. (Right) An example of the near term improvements that will be achieved in $\gamma$-ray searches for heavy DM. For this specific channel (DM$\rightarrow b\bar{b}$), it is clear that SWGO will considerably improve our reach, however, we note that for other channels leading results will be obtained by CTA. (We note that the results in this figure up to 2 TeV originally appeared in Ref. [128].)

## IV.    INDIRECT DETECTION

*Contributors: Carlos Blanco, J. Patrick Harding, Rebecca Leane, Elisa Pueschel, Nirmal Raj, Nicholas Rodd, Carsten Rott*

If dark matter is not perfectly stable, then it can decay and produce Standard Model states that stream through the Universe to our detectors. Similarly, it could be that the primary signature of DM appears through its annihilation into SM states.[2] Searching for DM through these final states is known as *indirect detection*, and should the DM fall in the ultraheavy mass window, the physics of these searches is considerably enriched.

For dark matter with mass well above the electroweak scale, the decays will inevitably produce a rich array of final states, including photons, neutrinos, and charged cosmic rays. This is true even if the DM decays only into neutrinos, as the neutrinos can shower electroweak bosons at such masses [123, 124]. Accordingly, such searches are inherently *multimessenger*, and benefit broadly from improvements in high energy astronomy. This point is highlighted in Fig. 3, where a partial set of present limits on the lifetime for DM$\rightarrow b\bar{b}$ are shown as an example. The results in green show limits obtained from low energy $\gamma$-rays collected by the *Fermi*-LAT telescope [125]. In Table I, we provide a list of current and future observatories and their sensitivities to the relevant standard model final states.

Note that while *Fermi* is optimized to search for $\mathcal{O}(\mathrm{GeV})$ photons, it can be sensitive to much heavier DM, as the high energy photons, electrons, and positrons produced in the decay will interact with cosmic background radiation, generating a cascade process converting energy down to lower scales [129–133]. The lower bound on the lifetime of heavy DM becomes essentially mass independent for DM masses above a few PeV if any appreciable portion of the mass energy is deposited into electromagnetic channels, since leptons and photons at these energies rapidly produce elec-

---

[2] See for example [121, 122]. for simplicity, we will mostly focus on decay in what follows.

| Experiment | Final state | Threshold/sensitivity | Field of view | Location |
|---|---|---|---|---|
| Current experiments | | | | |
| *Fermi* | Photons | $10$ MeV $- 10^3$ GeV | Wide | Space |
| HESS | Photons | 30 GeV - 100 TeV | Targeted | Namibia |
| VERITAS | Photons | 85 GeV - $> 30$ TeV | Targeted | USA |
| MAGIC | Photons | 30 GeV - 100 TeV | Targeted | Spain |
| HAWC | Photons | 300 GeV - >100 TeV | Wide | Mexico |
| LHAASO (partial) | Photons | 10 TeV - 10 PeV | Wide | China |
| KASCADE | Photons | 100 TeV - 10 PeV | Wide | Germany |
| KASCADE-Grande | Photons | 10 - 100 PeV | Wide | Italy |
| Pierre Auger Observatory | Photons | 1 - 10 EeV | Wide | Argentina |
| Telescope Array | Photons | 1 - 100 EeV | Wide | USA |
| IceCube | Neutrinos | 100 TeV - 100 EeV | Wide | Antarctica |
| ANITA | Neutrinos | EeV - ZeV | Wide | Antarctica |
| Pierre Auger Observatory | Neutrinos | 0.1 - 100 EeV | Wide | Argentina |
| Future experiments | | | | |
| CTA | Photons | 20 GeV - 300 TeV | Targeted | Chile & Spain |
| SWGO | Photons | 100 GeV - 1 PeV | Wide | South America |
| IceCube-Gen2 | Neutrinos | 10 TeV - 100 EeV | Wide | Antarctica |
| LHAASO (full) | Photons | 100 GeV - 10 PeV | Wide | China |
| KM3NeT | Neutrinos | 100 GeV - 10 PeV | Wide | Mediterranean Sea |
| POEMMA | Neutrinos | 20 PeV - 100 EeV | Wide | Space |

TABLE I. A non-exhaustive list of current and future indirect detection experiments sensitive to ultraheavy dark matter. See Refs. [134, 143–150].

tromagnetic cascades extending down to GeV energies which may become visible in the isotropic gamma-ray sky [133]. If the direct decay products can be observed, however, the constraints are generally stronger as seen from the other constraints in Fig. 3. The IceCube collaboration has set strong constraints on the prompt neutrinos produced by $\mathcal{O}$(PeV) DM [122, 126], and the results are shown in cyan in the plot. Finally, in red we show constraints from instruments searching for high energy cosmic rays, such as KASCADE, Pierre Auger Observatory, and Telescope Array [127, 134]. Many other instruments can search for the signature of heavy DM decay, including extensive air shower observatories such as HAWC [135] or Tibet AS$_\gamma$ [136, 137].

All indirect searches for heavy DM are underpinned by a detailed theoretical understanding of the production and propagation of the high energy particles involved. As mentioned above, the propagation effects are central in determining what spectrum of states arrives at the detector. When considering photon final states, a careful accounting of the inverse-Compton scattering (ICS) contribution is also required [138]. Furthermore, the full development of electromagnetic cascades, from a cycle of ICS and pair-production, must be taken into account in order to predict diffuse and isotropic signals [130–133, 139]. Before the propagation can be considered, however, a detailed understanding of the prompt spectra emerging from the initial decays is required. For the range of DM masses considered in this white paper, the center-of-mass energies involved in the decay can reach the Planck scale, well above energies involved in colliders such as the LHC. Nevertheless, a common approach adopted in the literature is to adapt simulation software optimized for the LHC, such as `Pythia` [140–142]. More recently, results have become available which perform dedicated calculations relevant to heavy DM decays. See in particular Ref. [124], where it is shown the spectra can depart significantly from `Pythia`. In the future, further work will be required to ensure that accurate predictions of how heavy DM should appear in our telescopes are available.

The current generation of ground-based imaging gamma-ray instruments (VERITAS, HESS and MAGIC) provide sensitivity to DM annihilation and decay up to and above 100 TeV [151, 152]. As mentioned previously for *Fermi*-LAT, while the energy sensitivity range of the current-generation instruments extends to ∼100 TeV, it is possible to probe DM masses well beyond this range, as the detected final-state photons from the DM decay or annihilation are expected at lower energies. The future CTA observatory, with sensitivity to gamma rays up to ∼300 TeV, will probe heavy DM with a factor of ∼10 better sensitivity than the current-generation instruments [153]. The future SWGO observatory, with energy reach up to >1 PeV, will be able to probe DM masses more than 100x better than the current HAWC constraints, as shown in Fig. 3 [128].

Neutrino observatories are also highly sensitive to decaying UHDM. Stringent bounds on heavy decaying DM have been achieved with IceCube excluding lifetimes of up to $10^{28}$ s depending on the decay mode [126, 154]. Bounds are extremely competitive to indirect searches with $\gamma$-rays [151] and are the world's strongest for DM masses above 10 TeV. These searches continue to explore BSM scenarios with DM masses beyond the reach of LHC and have a high discovery potential. Neutrino signals from heavy decaying DM have been discussed extensively [129, 155–160]. At present the observed astrophysical neutrino flux is not well enough measured to determine if it contains hints of heavy decaying DM [130, 156, 157, 161–167]. A significant increase in event statistics will be required to better constrain DM models or discover any signal. IceCube-Gen2 will be particularly important to obtain better sensitivity to heavy DM.

Mass-dependent limits may also be set on the DM annihilation cross sections in combination with scattering cross sections if we consider DM annihilations inside celestial bodies after they are captured. Annihilation products that may be detected are those that can escape the celestial body, such as neutrinos or long-lived mediators that can decay to visible states. Particularly strong constraints can come from DM capture in the Sun, by virtue of its large mass and proximity. Leading constraints on DM annihilations to neutrinos in the Sun come from Super-Kamiokande [168], IceCube [169], and ANTARES [170]. While these publications display limits for DM mass ≤ 10 TeV, these may be extended to higher masses in a straightforward manner, with a lower bound on the mass set only by the minimum detectable flux.

For heavy annihilating DM, more energetic neutrinos are produced, which leads to strong attenuation and a highly suppressed neutrino flux in the Sun. In this case, long-lived or boosted mediator production of neutrinos can greatly increase the detectability, as shown in Ref. [171]. For long-lived or boosted mediator production of gamma rays, it was pointed out in Ref. [171] that HAWC is an optimal observatory to search for heavy DM. HAWC has consequently set leading limits, displaying results for up to DM masses of $10^6$ TeV [172]. In the future, SWGO and LHAASO may have even better sensitivity to TeV-scale solar gamma rays from DM [171, 173]. **We strongly urge these collaborations to display the full extent of their limits along the DM mass axis**. The importance of populations of celestial bodies as the dominant source of DM annihilations, and as a probe of heavier-than-10-TeV DM with gamma rays, has also been investigated [174]. Similar DM gamma-ray searches using *Fermi*-LAT data of Jupiter have been performed [175]; although the results are only displayed up to 10 GeV DM mass, this search would also be able to provide sensitivity to much heavier DM.

A complementary method to probe heavy DM, down to smaller-than-electroweak scattering cross sections, is to observe the brightening of cold, isolated neutron stars (NS) via the transfer of DM kinetic energy during capture [176]. This may be done using upcoming infrared telescopes such as JWST, TMT and ELT [176] or telescopes operating at lower wavelengths if DM clusters into subhalos [177]; the possible presence of DM annihilations, a model-dependent issue, may boost the NS luminosity and help reduce telescope integration times. Various key particle and astrophysics implications of this probe have been investigated [178–198]. Assuming DM annihilation, this ce-

lestial body heating may also be detected using the Earth [199, 200] and exoplanets and brown dwarfs [201], which may be observable using JWST, Rubin, or the Roman telescopes in the next few years [201]. The observation of Population III stars is another probe of UHDM interactions with baryons [202, 203]. See also Refs. [204, 205]. Accumulation of asymmetric dark matter in compact astrophysical objects can also lead to low-mass black hole formation. Such black holes can be discovered by gravitational wave observatories, and this can in turn probe new parts of the parameter space [206].

## V. SUMMARY

Ultraheavy dark matter presents an exciting and relatively unexplored regime of possible dark matter candidates. A rich variety of production mechanisms and DM models are viable in this parameter space. Existing direct and indirect detection programs already exhibit significant sensitivity to a number of potential candidates. In the future, we encourage these experiments to display the constraints on the entire range of DM mass sensitivity up to the ultraheavy scales considered here. We have presented a few example searches in order to encourage other experimental collaborations to consider analyses of these heavy DM candidates. Moreover, a number of technologies in current development are quickly coming online and will continue to explore swathes of open parameter space. We look forward to continuing rapid developments in this exciting frontier in the search for dark matter.

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
