# Peer review of "Snowmass2021 Cosmic Frontier White Paper: Ultraheavy particle dark matter"

_SciPost Physics Core_

## Round 1 · Referee Report · Anonymous · 2023-3-7

Report

The acceptance criterion for SciPost Physics Core states that to be published, an article must "Detail one or more new research results significantly advancing current knowledge and understanding of the field." This submission seems to be a brief review article summarizing the current status of ultra heavy dark matter candidates. While useful, it is not clear if it contains any original results that have not appeared elsewhere. I am currently recommending rejection, but will be ready to reconsider if the authors can explain how this acceptance criterion is met.

---

## Round 1 · Referee Report · Anonymous · 2023-4-27

Report

The white paper reviews ultra-heavy dark matter with a mass below the Planck scale and is a good summary paper on the active research direction. It outlines several production mechanisms for ultraheavy dark matter: novel cosmological histories, gravitational production, primordial black holes, etc. Furthermore, heavy dark matter detection bounds from direct and indirect detection are presented. The article is a well-written review on this subject, and I would recommend it for publication.

Here I have some suggestions:

1) The detection sections focus on model-independent bounds on heavy dark matter, which looks decoupled from the production mechanism. Some discussion is given in the direct detection section but less in the indirection part. The paper could benefit greatly from more discussions on detecting ultra-heavy dark matter for several mechanisms presented in Sec 2.

2) The signatures of heavy dark matter models could also be detected by cosmological observations. The authors may consider adding some discussions on this.

3) For gravitational particle production through inflation, the authors wrote that the production is efficient when dark matter mass is comparable to the inflationary Hubble scale. But I think the production is efficient when the mass is much smaller than the Hubble scale.

---

## Editorial Decision

resubmitted